# Health Beliefs and Socioeconomic Determinants of COVID-19 Booster Vaccine Acceptance: An Indonesian Cross-Sectional Study

**DOI:** 10.3390/vaccines10050724

**Published:** 2022-05-05

**Authors:** Gede Benny Setia Wirawan, Ngakan Putu Anom Harjana, Nur Wulan Nugrahani, Pande Putu Januraga

**Affiliations:** 1Center for Public Health Innovation, Faculty of Medicine, Udayana University, Denpasar 80232, Indonesia; benny.wirawan007@gmail.com (G.B.S.W.); anomharjana@unud.ac.id (N.P.A.H.); nurwulannugrahani@gmail.com (N.W.N.); 2Department of Public Health and Preventive Medicine, Faculty of Medicine, Udayana University, Denpasar 80232, Indonesia

**Keywords:** COVID-19, vaccine, booster, vaccine hesitancy, health beliefs, trust, socioeconomic status

## Abstract

Introduction: The threat of new SARS-CoV-2 variants indicates the need to implement COVID-19 vaccine booster programs. The aim of this study was to identify the level of booster acceptance and its determinants. Methods: A cross-sectional online survey was conducted in Jakarta and Bali, Indonesia. Booster acceptance was divided into three categories: non-acceptor, planned acceptor, and actual acceptor. The primary independent variables were health beliefs, media influence, and trust in authoritative sources. Other covariates included demographics, socioeconomic status, and COVID-19 history. A primary analysis was conducted through multinomial logistic regression. The effects of the hypothetical situations on booster acceptance were tested using the Wilcoxon signed-rank test. Results: The final analysis included 2674 respondents with a booster acceptance rate of 56.3% (41.2% planned acceptors, 15.1% actual acceptors). Health beliefs, social media influence, and trust in authoritative information sources were identified as determinants for planned and actual booster acceptance. Socioeconomic status indicators were also identified as determinants for actual booster acceptance. Booster acceptance was increased in hypothetical scenarios involving booster requirements for work, travel, and accessing public places. Conclusions: Booster acceptance was found to be lower than the predicted primary vaccine acceptance prior to its launch. The acceleration of booster coverage requires strategies that leverage health beliefs and focus on people with a lower socioeconomic status.

## 1. Introduction

Two years of the COVID-19 pandemic has taught us that its epidemiology is characterized by waves of surging new daily cases. Each wave often corresponded with the emergence of a new SARS-CoV-2 variant taking over as the dominant circulating variant of the virus [1]. The latest wave the world experienced occurred around December 2021 to March of 2022 and was caused by the Omicron B.1.1.529.1 (BA.1) variant of SARS-CoV-2. This wave was characterized by the decoupling of the infection incidence rate from the hospitalization and mortality rates of COVID-19 [2].

Other than the increased accessibility of COVID-19 tests, this pattern can be attributed to the much higher COVID-19 vaccine coverage at this time [2,3]. At the beginning of the Omicron wave, around 50% of the world population had been fully vaccinated, compared to 6% at the beginning of the Delta wave [4,5]. In Indonesia, around 40% of the population had been fully vaccinated at the beginning of the local Omicron wave in early January 2022, although this figure was not evenly distributed across the vast archipelagic nation [6].

Despite this gain in the latest wave, the worldwide community must not be complacent. New SARS-CoV-2 variants may still emerge in the future. Currently, there is a risk posed by a subvariant of Omicron known as variant B.1.1.529.2 (BA.2), which may become the dominant variant in the near future [7]. Vaccines remain the best hope to mitigate the risk from Omicron BA.2 and other new variants. Early preliminary studies showed that antibodies elicited by complete vaccination as well as prior COVID-19 infection showed comparable neutralizing potential against the BA.1 and BA.2 variants [8,9]. However, there have been studies reporting lower neutralizing antibodies as early as 90 days after the administration of the second dose, which may lead to increased infection risk [10,11].

A booster dose has been proposed as a solution to maintain an effective level of immunity in the population [12]. Nevertheless, there are several barriers to the implementation of this strategy. On the basic level, there are potential issues of global vaccine supply leading to inequality in access to vaccines [12]. On the demand side, the level of acceptance for a COVID-19 vaccine booster dose remains unclear. Studies in China and the United Kingdom showed that booster acceptance varied between 75.2% and 91.1% [13,14,15]. However, real-life data showed a much slower uptake of the booster. Data from the United States showed that booster coverage only amounted to 29.6% of the population after more than 6 months since it first became available in August 2021 [4].

Indonesia officially initiated its booster campaign on 12 January 2022, complementing the earlier booster dose administration for healthcare workers in November 2021. By late March 2022, booster coverage reached 10.43% of the target population [6]. However, this figure is not distributed evenly across the archipelago. Jakarta and Bali can be seen as the benchmark for vaccine acceptance in Indonesia, being the provinces with highest full vaccination coverage [6]. These regions are prioritized for vaccination and booster campaigns due to their international renown as well their status as designated ports of entry to Indonesia in the national plan to open the border after the pandemic.

With vaccine hesitancy being a significant factor in COVID-19 vaccination campaigns, it cannot simply be assumed that the acceptance level of a booster dose would be equal to that of the first and second doses. Identifying the level of booster acceptance and its determinants in these regions can help stakeholders be more informed on the demand-side barriers of the COVID-19 vaccine booster campaign. Available evidence suggests that health beliefs and trust affect the intention to accept the primary COVID-19 vaccine doses [16]. Meanwhile, there has also been evidence showing that socioeconomic status affects primary COVID-19 vaccine acceptance [17]. To our knowledge, there has been little to no studies evaluating whether the same factors affect the acceptance of a booster dose. Thus, this study was conducted to evaluate whether health beliefs, trust, and socioeconomic status were the determinants affecting the acceptance of the COVID-19 vaccine booster dose.

## 2. Materials and Methods

### 2.1. Study Settings

Data were collected in the Jakarta and Bali provinces of Indonesia using an online survey platform. The data collection instrument was developed in December 2021 through to January 2022. Data collection was conducted during 6th to 16th February. The period in which this instrument was developed coincides with the development and initiation of the COVID-19 booster vaccine campaign in Indonesia, which officially kicked off on 12 January 2022. The booster doses administered were mostly heterologous, although the regulations allow for homologous booster doses in cases of inadequate supply [18]. Meanwhile, the data collection coincided with the initial increased daily cases of the Omicron prime wave.

At the time of the survey, national daily new COVID-19 cases averaged at around 40,000. In Jakarta and Bali, the locations of this survey, the daily new cases averaged at around 13,000 and 1700, respectively. At the same time, national COVID-19 vaccine coverage at the initiation of the data collection reached around 90% of the targeted population for the first dose and around 65% for the second dose, although there were some disparities between provinces. Jakarta and Bali were among the provinces with the highest vaccine coverage. At the beginning of the survey period, full COVID-19 vaccination coverage in Jakarta and Bali were 121% and 102% of the targeted population, respectively [19]. These figures were over 100% due to the large proportion of unrecorded internal migrants living in these predominantly urban provinces. As these people were not recorded in the regional demographic databases, they were not included in the calculation of the target population for the vaccine and booster campaign, although they are still eligible to obtain their shot in Jakarta and Bali.

### 2.2. Study Design

This study is of a cross-sectional analytic design, employing the 3Cs model of vaccine hesitancy as its theoretical framework. The 3Cs include confidence, complacency, and convenience [20]. Health beliefs and trust were the primary independent variables in this study. These concepts fit quite neatly under the 3Cs model, where perceived threat corresponds with complacency, perceived benefits and harms correspond with confidence, and perceived barriers correspond with convenience [21]. Trust, meanwhile, can be seen as a modifying factor that affect health beliefs.

Data collection was conducted in Jakarta and Bali by using an online survey with geolocation filtering. The minimum sample size for the proportion estimation was calculated to be 365, although there was an aim to include at least 2000 respondents in order to minimize biases from the online data collection [22]. The survey instrument was disseminated through a social media campaign. The campaign was conducted using paid ads on Twitter, Instagram, and Facebook. The inclusion criteria were residents of Jakarta and Bali, proven by the geolocation feature on the online survey platform, aged ≥18 years old, and had received at least one dose of the COVID-19 vaccine of any type. Respondents were given small monetary incentives amounting to IDR 50,000 (around USD 3) as compensation for communication fees incurred to participate in the study.

### 2.3. Measures

The primary variable of interest in this study was a respondent’s acceptance of a COVID-19 vaccine booster dose. This was measured via two questions: The first one asked whether the respondent had previously received a booster dose. Those who had already received a booster dose were categorized as ‘actual acceptors’. Secondly, for respondents who had not yet accepted a booster dose, they were asked about their likelihood of receiving it as soon as they would be eligible. The response to the second question was given in the form of a 5-itemed Likert scale ranging from ‘Would not accept’ to ‘Certainly would accept’. Those who answered ‘Certainly’ were categorized as ‘planned acceptors’ of a COVID-19 vaccine booster dose, while the others were categorized as ‘non-acceptors’. Further, non-acceptors were also asked about their acceptance of a booster dose in several hypothetical situations, including the emergence of new variants, a surge in new cases or mortality, and the implementation of hypothetical regulations.

The primary independent variables in the study were health beliefs, trust, and influence. The health beliefs measured included the perceived threat of COVID-19 as well as the perceived barriers, harms, and benefits of accepting a booster dose. These parameters were measured using items and a scoring system adapted from Chen et al. and Wong et al. [23,24]. Each belief was measured with a 6-itemed Likert scale, and each measure was displayed as an average score from a number of items (Appendix A). The scores were then converted into a scale of 0 to 10. Each health belief was then dichotomized, using the median as the cut-off point, into ‘high’ (≥median) and ‘low’ (<median) groups.

Trust and media influence were measured on the basis of the method reported by Wirawan et al. [16]. Trust in authoritative information sources was measured as a score from 5-itemed Likert scales ranging from ‘Not at all’ trusting the sources to ‘Strongly’ trusting them. Each item measured trust in institutions and officials of government, academia, and the healthcare industry. Meanwhile, the influence of media was measured as a function of trust and the frequency of exposure from a number of media, including print media, television, radio, online media, and social media. Trust was similarly measured in a 5-itemed Likert scale, while frequency was measured in a 6-itemed scale ranging from ‘Never’ to ‘Daily’, which correspond to scores of 0 and 5, respectively. The total influence score was a function of trust and frequency, with the score ranging from 0 to 25. The scores were then converted into a scale of 0 to 10. Each type of media was then dichotomized, using the median as the cut-off point, into ‘high’ (≥median) and ‘low’ (<median) groups.

Other factors measured in this study included sociodemographic data and histories of COVID-19 infection. The sociodemographic factors measured included age, sex, religion, education level, employment status, monthly income, and health insurance. Meanwhile, respondents were also asked about the COVID-19 infection history of themselves as well as their family and/or close friends. This included histories of infection, hospitalization, and mortality.

### 2.4. Analysis

Proportions were presented as raw figures as well as percentages. Ordinal scores from Likert scales were presented as median and interquartile ranges (IQRs). Primary significance testing was conducted using multivariate multinomial logistic regression to identify the factors associated with the planned and actualized acceptance of a COVID-19 vaccine booster dose. The Wilcoxon signed-rank test was also conducted to test the change in the acceptance of a booster dose in each hypothetical scenario. All the analyses were conducted using IBM SPSS 23.0 (IBM Corp., Armonk, NY, USA).

### 2.5. Ethical Consideration

The study protocol has been reviewed and approved by the appropriate ethical committee, with the approval letter no. 0032T/III/LPPM-PM.10.05/12/2021. All respondents were provided with information regarding the study and given the opportunity to communicate directly with the study team. All respondents provided their informed consent to participate in the study.

## 3. Results

The online survey garnered 3088 responses. After cleaning in addition to removing duplicate and noneligible entries, the final analyses included 2674 respondents, with their characteristics detailed in Table 1. This figure included 1102 people who planned to accept a booster dose and 403 people who had already accepted one. Among those who had not had their booster dose, the overwhelming majority had been completely vaccinated (85.6% in the non-acceptor group and 95.0% among the planned acceptors). For the first and second doses, the majority (61.1%) received inactivated virus vaccines, while 29.4% received viral vector vaccines and 8.0% received mRNA vaccines. There were 1.5% of respondents who received other types of vaccines or did not know the types of vaccines they received.

Females made up 58% of the respondents, while 70.8% of the respondents were detected to be from Jakarta. The overall median age was 29 (IQR of 24–35) years old, and the median ages for the non-acceptor, planned acceptor, and actual acceptor groups were 29 (IQR of 23–35), 29 (IQR of 24–35), and 29 (IQR of 25–36) years old, respectively. From a socioeconomic perspective, 52.8% of the overall respondents were high school graduates. However, only half of the respondents were found to be working, with 32.5% working full time and 18.3% working part time, with 30.8% of respondents making less than IDR 1 million (around USD 80) per month. Around one-third (33.1%) of the respondents were covered by subsidized public health insurance.

The COVID-19 infection history of the respondents showed that a large majority (88.6%) reported that they had never been infected. Only 2.3% reported to have been infected and hospitalized with COVID-19. The majority of respondents (56.9%) also reported to never have had their family members or close friends infected with COVID-19. However, 10.2% reported to have had someone hospitalized with COVID-19, and 16.2% reported COVID-19-related mortality among their family and friends.

The overall media influence (Table 2) was dominated by online media, social media, and television, which received a median score of 4.80 (IQR of 3.60–6.40) in a scale of 0 to 10. Meanwhile, the median perceived benefits and perceived harms scores indicate that the average respondent believes that a booster dose is at least somewhat beneficial while not posing significant health risks.

Multinomial logistic regression (Table 3) showed that health beliefs and trust were significantly associated with being a planned or actual acceptor of a COVID-19 vaccine booster dose. As expected, perceived threats and benefits were positively associated with both planned and actual booster acceptance. Meanwhile, perceived barriers and harms were negatively associated with both planned and actual booster acceptance. Trust in authoritative information sources was also positively associated with both planned and actual booster acceptance. However, while social media was found to be positively associated with both planned and actual booster acceptance, the influence of print media was positively associated with planned acceptance, while the influence of television was negatively associated with actual booster acceptance.

A further analysis of potential situations to increase booster acceptance (Figure 1) found that the worsening of the pandemic situation, such as the emergence of a new COVID-19 variant or a surge in new cases, actually decreased acceptance compared to the baseline. Instead, booster acceptance increased in hypothetical scenarios that involved changes in policy that mandated a booster dose for several activities, such as to work, travel, or access public places.

## 4. Discussion

The results identified that the overall booster acceptance among respondents was 56.3%, which includes 41.2% planned acceptors and 15.1% actual acceptors who mostly received mRNA vaccines. This acceptance level is considerably lower when compared to prior studies regarding vaccine acceptance immediately prior to its launch, which reached around 70% [16,25]. Furthermore, the data reported acceptance in Jakarta and Bali, which is an overestimation of the national booster acceptance level. At the time of data collection, only around 3.5% of the targeted population nationwide had received a booster dose. This is also in line with what is known of the COVID-19 vaccine acceptance rate in Jakarta and Bali compared to the national average [6]. This figure is further inflated by the fact that 26.8% of the actual acceptors received their shot in November 2021, in the booster scheme for healthcare workers. This figure grossly overrepresents the proportion of healthcare workers in the population. In short, the actual booster acceptance rate at the national level may actually be even lower than what is reported here.

Pandemic fatigue could be attributed as one of the reasons leading to the lower acceptance of a booster dose. After two years of continuously being exposed to news about the pandemic, some people began to tune it out. This was exacerbated by repeated unfulfilled hopes regarding COVID-19 mitigation strategies (e.g., ‘Two weeks to flatten the curve’, ‘Once we’re all vaccinated, we can go back to normal’), which may have led to declining trust and unwillingness to adhere to new recommended preventive behaviors, such as taking a booster dose [26,27].

Moreover, complacency regarding the threat of COVID-19 may have also crept into Indonesian society. The latest wave of COVID-19 caused by the Omicron variant did not come with high hospitalization or mortality rates, especially among the younger age group [2]. This decoupling between infection, hospitalization, and mortality for Omicron BA.1 may have led to a false sense of security and reduced perceived threat. Considering the median age (29 years old), lower perceived threat may be prevalent among the respondents.

Pandemic news has also been less prevalent in the Indonesian news cycle, taken over by other economic, social, and political issues. This is further enforced by a lax response from the government, especially compared to the strict social restrictions imposed during the Delta wave [28]. Combined, these factors may have led to complacency and lower perceived threat.

A primary analysis identified health beliefs as a consistent determinant of planned and actual booster acceptance. Predictably, perceived harms and benefits increased the likelihood of acceptance, while perceived barriers and harms reduced it. However, it is interesting to see that socioeconomic factors were found as significant and independent determinants only for actual booster acceptance but not for planned acceptance. Socioeconomic factors identified as determinants include employment, education level, income, and health insurance coverage. The role of socioeconomic status is especially strong in Bali (Appendix A). This is the one of the first times that a study has identified the socioeconomic-based inequality of COVID-19 vaccine coverage in Indonesia.

It is interesting to see the lack of association found between histories of COVID-19 infection, both for self and family/friends, with booster acceptance. This is contradictory to previous finding in the UAE and the USA [29,30]. However, this finding is consistent with a previous Indonesian study [16]. A history of infection could conceivably increase one’s perceived threat. As such, the effect of prior infections may be subsumed under this variable. More interestingly, a very low level of respondents reporting a prior history of COVID-19 infection is seen, either to themselves or to their family and friends. While this is in line with the number of cumulative COVID-19 cases in Indonesia (6 million cases in a country of 280 million), it can also be attributed to the reluctance to report prior infection histories in addition to low testing numbers, especially early in the pandemic [31,32].

The findings also show that the importance of perceived threat is further demonstrated by the determinants of booster acceptance identified in this study. Health beliefs and trust in authoritative information sources were found to be independent determinants for planned and actual booster acceptance in multinomial logistic regression. This is in line with previous studies on COVID-19 vaccine acceptance and more recent studies on booster acceptance, both in Indonesia and elsewhere in the world [33,34]. Moreover, religion was also identified as a factor in determining booster acceptance. All these pointed to the role of one’s belief system as a determinant of COVID-19 vaccine hesitancy. This is also in line with previous findings on vaccine hesitancy in Indonesia [35,36].

The role of beliefs and trust in booster acceptance mimics prior findings regarding vaccine acceptance in Indonesia. Conspiracy beliefs and trust were identified as independent determinants for vaccine acceptance [16]. COVID-19 conspiracies mostly revolved around the level of threat of the disease and harms posed by the vaccines [37,38]. Meanwhile, Islamic religion added another layer of concern regarding the halal status of the vaccine, which persisted despite reassurances by the supreme religious body in Indonesia, the Indonesian Ulema Council [39].

Beliefs, including health beliefs, are considerably influenced by sources of information and the media. From these results, it can be seen that social media is the most influential media, and that its influence is positively associated with booster acceptance. This is in line with previous findings from Indonesia [16]; there have been previous reports linking social media engagement with lower vaccine hesitancy. However, this was conditional on the types and sources of social media content consumed. Engagement with news and authoritative sources on social media is associated with lower vaccine hesitancy [40].

Regarding the role of socioeconomic status as a determinant for booster acceptance, this is in line with previous findings on determinants of COVID-19 preventive behavior, including on vaccine acceptance [17,41]. Many factors can contribute here. Different rates of health literacy related to socioeconomic status may affect health beliefs and, subsequently, attitudes toward a booster dose [17]. People with a lower socioeconomic status and dependent on day-to-day income may be reluctant to take a booster dose due to concern about losing potential income as they suffer from potential adverse effects [42]. Furthermore, the fact that stay-at-home-wives were found to have the lowest likelihood of having received a booster dose may also show the role of gender inequality, a known factor in healthcare utilization among women [43].

Government policies also play a role, as they pushed for booster administration for the workforce in order to support economic productivity [44]. Despite regulations prioritizing booster shots for vulnerable groups, such as the elderly and people with comorbidities, publicly organized mass booster vaccination events were held mostly for civil servants, employees, and other general population groups [45,46]. In effect, these factors led to preferential access for a booster dose being given to people in the productive age group, especially those employed in the formal sector. The especially strong effect of full-time employment and having unsubsidized public health insurance (hallmarks of formal sector employees) in Bali lend further credence to this theory. The hospitality industry, which employs most of Balinese workforce, has higher incentives to require boosters for their employees to keep up with safety standards.

These results pointed to the need for an acceleration of COVID-19 booster coverage in Indonesia to mitigate the risk of future waves. The findings support this assertion as respondents were more likely to accept booster dose if it were to be mandated by the government. People with a lower socioeconomic status should be given special attention in accelerating booster coverage. This could be achieved by both ‘pull’ and ‘push’ strategies. Pull strategies can be achieved by health promotion, increasing health literacy among people with a lower socioeconomic status and targeted at promoting health beliefs associated with booster acceptance. Well-designed social media campaigns seem to be the most cost-effective way to conduct this campaign, owing to their influence and low cost barrier. Meanwhile, ‘push’ strategies could be achieved by implementing booster requirements to work or access public places. This strategy is also aimed at reducing the risk of symptomatic infection due to viral exposure to inadequately vaccinated individuals.

Further strategies could also leverage Indonesian local communities. As a traditionally collectivist society, Indonesia has robust social infrastructures in local communities that can help organize public policies, including vaccination. The collectivist nature of Indonesian culture has also led to the role of perceived social norms in promoting certain behaviors [47], which would be leveraged in this scheme. Social infrastructures, such as local villages or traditional institutions, can be leveraged by health authorities to promote booster acceptance. Similar schemes have previously been implemented in the initial COVID-19 vaccination campaign [48] but have yet to be fully implemented for the booster dose campaign.

These social approaches should not neglect the build-up of the health infrastructure necessary for vaccination, however. There is an issue of the supply of mRNA and viral vector vaccines, as they were mostly received through COVAX scheme, whereas inactivated virus vaccines have been able to be produced domestically [49]. This supply imbalance itself plays a role in the heterologous booster policy. The more-abundant inactivated virus vaccines were prioritized for the first and second doses as the government attempted to catch up with regional vaccine coverage disparities. However, the policy of using mRNA and viral vector vaccines puts a huge strain on the Indonesian cold chain capacity for the booster campaign, which was severely inadequate for Indonesia’s large and diverse geography [49]. These issues need to be addressed as the booster campaign pushes further into more rural areas.

This study is one of the first reporting the level of booster acceptance and its determinants in Indonesia. It is also among the first to report the inequality of vaccine coverage based on socioeconomic status in Indonesia. However, this study is not without its limitations. The main limitation of this study is its data collection method using an online survey. While robust and efficient, this data collection method has an inherent flaw that risks selection bias and careless responses [22]. The selection bias in this study led to the overrepresentation of a younger age group and people with higher education compared to the actual demographics of Jakarta and Bali, although it is in line with the demographics of internet users in Indonesia [50]. This demographic difference was not considered statistically and may constitute a limitation of this study. Moreover, Indonesia is a vast and diverse country, yet the data collection was conducted specifically for Jakarta and Bali, which may further overestimate booster acceptance. This demographic limitation requires more careful inference from the data presented in this study to the national level.

## 5. Conclusions

The acceptance of a booster dose was found to be lower compared to vaccine acceptance prior to its launch in 2021. Health beliefs were found to be the primary determinant for the intention to receive a booster, while the actual booster acceptance was also affected by socioeconomic factors. These results indicate the need for more aggressive campaigns to increase acceptance. This can be achieved by health promotion through leveraging health beliefs associated with booster acceptance, with special attention to people with a lower socioeconomic status. The implementation of policies requiring a booster dose for work, travel, and access to public places was also indicated to be able to increase booster acceptance.

## Figures and Tables

**Figure 1 vaccines-10-00724-f001:**
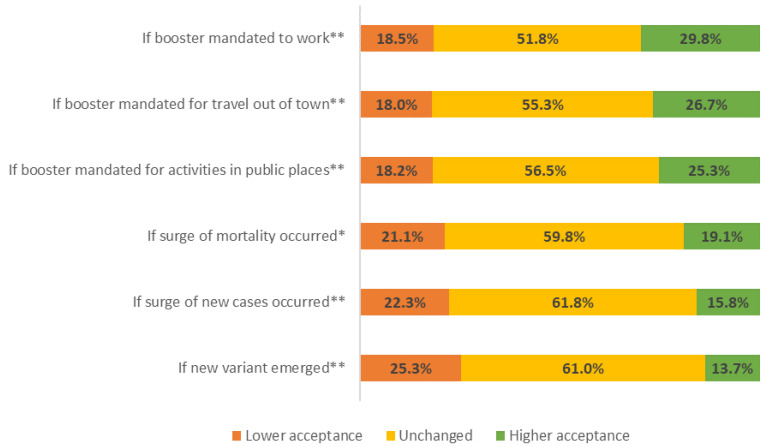
Changes to COVID-19 vaccine booster acceptance in hypothetical scenarios among initial non-acceptors (*n* = 1169). Significance of the changes compared to baseline acceptance were tested using the Wilcoxon signed-rank test (* *p* < 0.05; ** *p* < 0.01).

**Table 1 vaccines-10-00724-t001:** Demographic, COVID-19 infection, and vaccination history characteristics of respondents.

Variables	Total	COVID-19 Booster Vaccine Acceptance		
	(*n* = 2674)	Non-Acceptor	Planned Acceptor	Actual Acceptor
		(*n* = 1169)	(*n* = 1102)	(*n* = 403)
Demographics				
Sex, *n* (%)				
Male	1123 (42.0)	502 (42.9)	427 (38.7)	194 (48.1)
Female	1551 (58.0)	667 (57.1)	675 (61.3)	209 (51.9)
Age (years), median (IQR)	29 (24–35)	29 (23–35)	29 (24–35)	29 (25–36)
Location, *n* (%)				
Jakarta	1894 (70.8)	931 (79.6)	797 (72.3)	166 (41.2)
Bali	780 (29.2)	238 (20.4)	305 (27.7)	237 (58.8)
Religion, *n* (%)				
Islam	1910 (71.4)	978 (83.7)	778 (70.6)	154 (38.2)
Non-Islam	764 (28.6)	191 (16.3)	324 (29.4)	249 (61.8)
Education, *n* (%)				
Not completed high school	222 (8.3)	139 (11.9)	75 (6.8)	8 (2.0)
Completed high school	1412 (52.8)	725 (62.0)	596 (54.1)	91 (22.6)
Completed college	1040 (38.9)	305 (26.1)	431 (39.1)	304 (75.4)
Health insurance, *n* (%)				
Subsidized public insurance	894 (33.1)	460 (39.3)	367 (33.3)	57 (14.1)
Unsubsidized public insurance	994 (37.2)	339 (29.0)	427 (38.7)	228 (56.6)
Private insurance	796 (29.8)	370 (31.7)	308 (27.9)	118 (29.3)
Employment, *n* (%)				
Unemployed	339 (12.7)	184 (15.7)	126 (11.4)	29 (7.2)
Stay-at-home wife	657 (24.6)	351 (30.0)	285 (25.9)	21 (5.2)
Student	318 (11.9)	149 (12.7)	132 (12.0)	37 (9.2)
Part-time employment	490 (18.3)	213 (18.2)	208 (18.9)	69 (17.1)
Full-time employment	870 (32.5)	272 (23.3)	351 (31.9)	247 (61.3)
Monthly income, *n* (%)				
<IDR million	823 (30.8)	434 (37.1)	317 (28.8)	72 (17.9)
IDR 1 million–IDR 3 million	752 (28.1)	330 (28.2)	310 (28.1)	112 (27.8)
IDR 3 million–IDR 5 million	674 (25.2)	295 (25.2)	281 (25.5)	98 (24.3)
>IDR 5 million	425 (15.9)	110 (9.4)	194 (17.6)	121 (30)
**COVID-19 Vaccination History**				
Vaccination history, *n* (%)				
1 dose	223 (8.3)	168 (14.4)	55 (5.0)	N/A
2 doses	2048 (76.6)	1001 (85.6)	1047 (95.0)	N/A
3 doses	403 (15.1)	N/A	N/A	403 (100.0)
Vaccine type (1st and 2nd dose), *n* (%)				
Inactivated virus	1635 (61.1)	725 (62.0)	602 (54.6)	308 (76.4)
Viral vector	786 (29.4)	320 (27.4)	415 (37.7)	51 (12.7)
mRNA vaccine	213 (8.0)	99 (8.5)	77 (7.0)	37 (9.2)
Other or do not know	40 (1.5)	25 (2.1)	8 (0.7)	7 (1.7)
Booster scheme accepted, *n* (%)				
Booster for health worker	N/A	N/A	N/A	108 (26.8)
Booster for general public				295 (73.2)
Vaccine type (booster dose), *n* (%)	N/A	N/A	N/A	
Inactivated virus				13 (3.2)
Viral vector				95 (23.6)
mRNA vaccine				286 (71.0)
Other or do not know				9 (2.2)
COVID-19 Infection History				
Infection history, *n* (%)				
Never been infected	2370 (88.6)	1050 (89.8)	963 (87.4)	357 (88.6)
Infected, but never hospitalized	242 (9.1)	91 (7.8)	115 (10.4)	36 (8.9)
Infected and hospitalized	62 (2.3)	28 (2.4)	24 (2.2)	10 (2.5)
Infection history family/friends, *n* (%)				
No infection	1521 (56.9)	745 (63.7)	598 (54.3)	178 (44.2)
Infection only	440 (16.5)	170 (14.5)	178 (16.2)	92 (22.8)
Hospitalization	281 (10.5)	108 (9.2)	124 (11.3)	49 (12.2)
Mortality	432 (16.2)	146 (12.5)	202 (18.3)	84 (20.8)

**Table 2 vaccines-10-00724-t002:** Media influence, trust, and health belief scores (range from 0 to 10) among respondents.

Variables	Total	COVID-19 Booster Vaccine Acceptance		
	(*n* = 2674)	Non-Acceptor	Planned Acceptor	Actual Acceptor
		(*n* = 1169)	(*n* = 1102)	(*n* = 403)
Media Influence				
Media influence, median (IQR)				
Influence of print media	4.80 (1.60–6.40)	3.60 (1.20–4.80)	4.80 (2.40–6.40)	4.00 (1.20–6.40)
Influence of television	4.80 (3.60–6.40)	4.80 (3.20–6.40)	6.40 (4.80–8.00)	4.80 (3.20–6.40)
Influence of radio	3.20 (0.00–6.00)	2.40 (0.00–4.80)	4.80 (0.80–6.40)	3.20 (0.00–4.80)
Influence of online media	4.80 (3.60–6.40)	4.80 (3.60–6.40)	6.40 (4.80–8.00)	4.80 (3.60–6.40)
Influence of social media	4.80 (3.60–6.40)	4.80 (3.20–6.40)	6.40 (4.80–8.00)	4.80 (3.20–6.40)
**Trust**				
Trust in authoritative sources, median (IQR)	7.50 (5.83–8.33)	7.08 (5.00–7.50)	7.50 (7.08–9.18)	7.50 (6.68–8.75)
**Health Beliefs**				
Perceived threat, median (IQR)	6.46 (4.46–8.00)	5.74 (3.80–7.46)	7.10 (5.14–8.60)	7.40 (5.46–8.40)
Perceived barriers, median (IQR)	3.00 (0.66–5.34)	4.00 (2.00–6.00)	2.00 (0.00–5.00)	1.66 (0.00–3.34)
Perceived harms, median (IQR)	2.00 (0.40–4.00)	3.20 (1.60–4.80)	1.20 (0.00–2.80)	1.20 (0.00–2.80)
Perceived benefits, median (IQR)	8.00 (6.40–9.60)	6.80 (5.60–8.00)	8.80 (8.00–10.00)	8.00 (7.20–9.60)

**Table 3 vaccines-10-00724-t003:** Multinomial logistic regression model for COVID-19 vaccine booster acceptance.

Variables	Planned to Accept	Already Accepted
	aOR (95% CI)	aOR (95% CI)
Health Belief		
Perceived threat		
Low	1	1
High	1.69 (1.38–2.08) **	2.33 (1.73–3.14) **
Perceived barriers		
Low	1	1
High	0.65 (0.52–0.81) **	0.31 (0.23–0.43) **
Perceived harms		
Low	1	1
High	0.47 (0.38–0.59) **	0.47 (0.34–0.64) **
Perceived benefits		
Low	1	1
High	2.81 (2.27–3.49) **	1.85 (1.35–2.54) **
**Media Influence and Trust**		
Influence of print media		
Low	1	1
High	1.51 (1.19–1.93) **	1.35 (0.96–1.90)
Influence of television		
Low	1	1
High	1.15 (0.86–1.53)	0.65 (0.44–0.98) *
Influence of radio		
Low	1	1
High	0.95 (0.75–1.20)	1.06 (0.76–1.48)
Influence of online media		
Low	1	1
High	0.95 (0.69–1.31)	0.94 (0.60–1.47)
Influence of social media		
Low	1	1
High	1.64 (1.24–2.18) **	1.69 (1.14–2.50) **
Trust in authoritative sources		
Low	1	1
High	1.45 (1.16–1.81) **	1.20 (0.87–1.66)
**Demographics and History**		
Sex		
Male	1	1
Female	1.18 (0.93–1.49)	1.00 (0.73–1.37)
Age (per incremental years)	1.00 (0.99–1.01)	1.02 (1.00–1.04) *
Location		
Jakarta	1	1
Bali	1.07 (0.80–1.43)	2.31 (1.59–3.36) **
Religion		
Islam	1	1
Non-Islam	2.19 (1.62–2.98) **	3.22 (2.22–4.67) **
Education		
Not completed high school	0.78 (0.55–1.11)	0.76 (0.34–1.71)
Completed high school	1	1
Completed college	1.43 (1.12–1.82) **	3.29 (2.31–4.70) **
Employment		
Full-time employment	1	1
Part-time employment	0.81 (0.60–1.09)	0.47 (0.32–0.71) **
Student	0.76 (0.50–1.17)	0.36 (0.19–0.67) **
Stay-at-home wife	0.75 (0.54–1.06)	0.14 (0.08–0.26) **
Unemployed	0.66 (0.46–0.96)	0.22 (0.12–0.38) **
Monthly income		
<IDR 1 million	1	1
IDR 1 million–IDR 3 million	1.08 (0.82–1.43)	0.75 (0.46–1.22)
IDR 3 million–IDR 5 million	1.03 (0.75–1.40)	0.65 (0.38–1.10)
IDR 5 million	1.58 (1.07–2.33) *	1.20 (0.67–2.16)
Health insurance		
Subsidized public insurance	1	1
Unsubsidized public insurance	1.17 (0.92–1.49)	2.24 (1.52–3.30) **
Private insurance	0.87 (0.68–2.33)	1.22 (0.81–1.84)
COVID-19 infection history		
Never infected	1	1
Infected, never hospitalized	1.20 (0.85–1.70)	0.76 (0.46–1.23)
Infected and hospitalized	0.67 (0.34–1.31)	0.60 (0.24–1.51)
COVID-19 history family/friends		
No infection	1	1
Infection only	1.15 (0.86–1.54)	1.31 (0.89–1.94)
Hospitalization	1.11 (0.79–1.56)	1.12 (0.70–1.79)
Mortality	1.24 (0.93–1.66)	1.49 (1.00–2.22)

* *p* < 0.05; ** *p* < 0.01.

## Data Availability

The data presented in this study are available on reasonable request from the corresponding author.

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
