# Peer review of "Health Beliefs and Socioeconomic Determinants of COVID-19 Booster Vaccine Acceptance: An Indonesian Cross-Sectional Study"

_vaccines, 2022, doi:10.3390/vaccines10050724_

Round 1

Reviewer 1 Report

This is an interesting study regarding the identification of the level of booster acceptance and its determinants during the covid-19 pandemic. The authors collected a large amount of dataset using the indicative methodology focused on the Indonesian Population. However, in my opinion the paper has some shortcomings in regards to some data presentation and text and I feel that this collected dataset has not been utilized to its fulfill extent.

Below I provided remarks:

Major recommendations:

  1. lack of specific research questions.
  2. the result section of the research paper must be re-structured. Descriptive analysis must be followed by the graphs results and the tables.

3.a short explanation of the findings could be included to the “Results” section.

Minor recommendations:

  1. “Materials and Methods” section is well described, although study limitations did not refer to.
  2. table 2 did not refer in the text
  3. the results' presentation structure must be revised. Using of graphs instead of table will make the results to be more clear. 

Author Response

We would like to convey our gratitude to the editorial board and all reviewers who have taken the time to review and provide comments for our manuscript. We found the comments helpful in improving our writing and has consequently made some revisions accordingly which we will enumerate here.

Reviewer #1

  1. lack of specific research questions.

Response:

Thank you for this comment. We re-check the introduction section and indeed we did not provide a specific research question although we did provide the aims of the study. Accordingly, we have added a paragraph detailing our research questions and the reasoning behind the choice of variables analyzed in this study. Please refer to lines 77-84.

  1. the result section of the research paper must be re-structured. Descriptive analysis must be followed by the graphs results and the tables.

the results' presentation structure must be revised. Using of graphs instead of table will make the results to be more clear.

Response:

Thank you for the input. We took this comment to heart and has accordingly convert some data presentation into graph. We substituted Table 4 into Figure 1 which essentially convey the same information. However, due to the volume of data presented in Table 2 and 3 we were not able to convert them into graphs or plots condense enough for the manuscript. However, we did convert the scores used in Table 2 into a uniform scale of 0 to 10 to make it easier to understand. Please refer to the changes in Table 2 and Figure 1.

  1. a short explanation of the findings could be included to the “Results” section.

Response:

Thank you for your comment. We agree that summarized explanation of the findings is necessary to provide context for further discussion. We did provide the narrative readings of all tables and figures provided in the “Results” section. We also provided condensed and summarized readings of the results in the “Discussion” section at the beginning of each topic of discussion. Please refer to lines 238-240, lines 270-278, and lines 335-338.

  1. “Materials and Methods” section is well described, although study limitations did not refer to.

Response:

Thank you for the comment. We agree that transparent reporting of the study limitations is important in academia. We did provide an assessment of the limitations of our study in the last paragraph of the discussion section. Please refer to lines 367-378.

  1. table 2 did not refer in the text

Response:

Thank you for noticing this shortcoming. We have revised the text accordingly by adding a reference to Table 2 in the narrative text. Please refer to line 209.

Reviewer 2 Report

  • I dislike use of 1st person in scientific papers as it introduces a concept of personal bias that should be avoided. "We" is used in the abstract and as used elsewhere as per the introduction from which the following quotation has been taken, “ …we must not be complacent…” or “…Currently, we are posed by the risk of…” or “…Vaccines remain our best hope to mitigate…”
  • There should not be commas before conjunctives, such as 'and' and 'but', as appear repeatedly in the abstract and eslewhwere.
  • The following sentence, found in the conclusions section of the abstract is confusing and should be rewritten to make sense, "  Booster acceptance was found lower than predicted pri-23 mary vaccine acceptance prior to its launch.”
  • Reading the Introduction reveals poor grammar and syntax which needs to be corrected as it makes reading of the paper difficult. I offer the following quotations as a perfect example, “ … At the beginning of the Omicron wave, around 50% of the world’s population has been fully vaccinated, compared to 6%, at the beginning of the Delta wave [4,5]. In Indonesia, around 40% of the people have been fully vaccinated at the beginning of the local Omicron wave…” I have added the corrections in red to show how it should have read.
  • Words like ‘However” are superfluous and should be omitted as they add nothing to the context of the paper, as per the following quotation taken form the introduction section, “…However, there has been studies…”. This same quotation also highlights grammatical errors as ‘studies’ are pleural and ‘has is singular and should have been “have”.
  • Based on these comments alone, I feel that the paper needs English language editing before it is submitted for scientific evaluation as reading it is too frustrating for a pedant such as me. I would be prepared to look at the paper again, once the English has been improved, because I believe it has an important message and may be very suitable for publication but, at the moment, the reading of this submission is too demanding, for the reasons set out above.

Author Response

We would like to convey our gratitude to the editorial board and all reviewers who have taken the time to review and provide comments for our manuscript. We found the comments helpful in improving our writing and has consequently made some revisions accordingly which we will enumerate here.

Reviewer #2

  1. Based on these comments alone, I feel that the paper needs English language editing before it is submitted for scientific evaluation as reading it is too frustrating for a pedant such as me. I would be prepared to look at the paper again, once the English has been improved, because I believe it has an important message and may be very suitable for publication but, at the moment, the reading of this submission is too demanding, for the reasons set out above.

Response:

Thank you for your commentary about the quality of the English language in our original manuscript. Adequately understandable English is indeed a requirement for a good scientific publication, and we took this comment to heart. Consequently, we have put the manuscript through a professional English editing service provided by MDPI (certificate attached). I hope the edited manuscript would be understandable enough to progress to a review of its content.

Round 2

Reviewer 2 Report

While the text ias not as properly English edited as I would like, still containing superfluous words , such as 'however', and still using commas before conjunctives, such as 'and', it is far easier to read and hence I believe it is almost acceptable. I would love to see btter English editing, even recognising that there has been in house language editing, I consider the paper suitable for publication.